# The Dead Salmons of AI Interpretability

## Abstract

In a striking neuroscience study, the authors placed a dead salmon in an MRI scanner and showed it images of humans in social situations. Astonishingly, standard analyses of the time reported brain regions *predictive* of social emotions. The explanation, of course, was not supernatural cognition but a cautionary tale about misapplied statistical inference. In AI interpretability, reports of similar "dead salmon" artifacts abound: feature attribution, probing, sparse auto-encoding, and even causal analyses can produce plausible-looking explanations for randomly initialized neural networks. In this work, we examine this phenomenon and argue for a pragmatic **statistical–causal reframing**: explanations of computational systems should be treated as parameters of a (statistical) model, inferred from computational traces. This perspective goes beyond simply measuring statistical variability of explanations due to finite sampling of input data; interpretability methods become statistical estimators, and findings should be tested against explicit and meaningful **alternative computational hypotheses**, with uncertainty quantified with respect to the postulated statistical model. It also highlights important theoretical issues, such as the identifiability of common interpretability queries, which we argue is critical to understand the field's susceptibility to false discoveries, poor generalizability, and high variance.

## 1. Introduction

In 2009, researchers placed a dead salmon in an MRI scanner, showed it photographs of humans in social situations, and ostensibly asked it to judge their emotions (Bennett et al., 2009). Standard analysis pipelines commonly used at

[1]Anonymous Institution, Anonymous City, Anonymous Region, Anonymous Country. Correspondence to: Anonymous Author <anon.email@domain.com>.

Preliminary work. Under review by the International Conference on Machine Learning (ICML). Do not distribute.

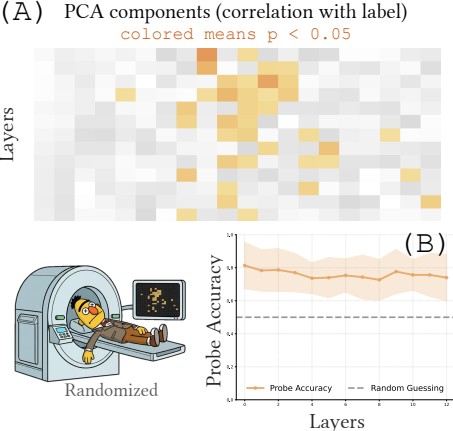

*Figure 1.* **Minimal dead salmon artifacts**. We extract token representations from a randomly initialized BERT for 300 IMDb sentences and average over sequence length. (A) Several principal components are spuriously correlated with sentiment labels. (B) A simple probe trained on layer representations achieves nontrivial cross-validated accuracy.

the time surprisingly returned brain voxels as significantly predictive of emotional situations. The error arose from a failure to correct for multiple comparisons within the statistical analysis pipeline. The "dead salmon" demonstration of false positives contributed to a larger reckoning in the field of neuroscience. For instance, an influential study showed that different research groups obtained different results even when analyzing the same dataset and the same research question (Botvinik-Nezer et al., 2020). Subsequent work identified several sources of *statistical fragility*. Widely used statistical procedures embedded in standard analysis pipelines were shown to inflate false-positive rates (Eklund et al., 2016), an effect worsened by non-independent analyses producing spuriously large brain–behavior correlations (Vul et al., 2009). Also, early neuroimaging research was constrained by small samples and limited data availability (Button et al., 2013; Marek et al., 2022), exacerbating overfitting and spurious associations. Moreover, fMRI had been criticized for offering predictive explanations, rather than functional ones, resulting in little clinical relevance (Lyon, 2017). Finally, reverse inference emerged as a central interpretative problem, given that individual neural systems are not uniquely associated with specific cognitive functions (Poldrack, 2006; Duncan & Owen, 2000).

AI interpretability now faces its own *dead salmon* issues, similarly begging for a larger reevaluation of its statistical foundations. A growing body of work has shown that many influential methods, including feature attribution (Adebayo et al., 2018), probing classifiers (Ravichander et al., 2021), sparse autoencoders (Heap et al., 2025), circuit discoveries (Méloux et al., 2025), and causal abstractions (Sutter et al., 2025), can yield plausible-looking explanations even when applied to **random neural networks**. In Figure 1, we report a minimum dead salmon artifact from analyzing activations of a fully randomized BERT model in a sentiment analysis task where both correlation analysis and probing find highly significant *explanations*. Such striking failure modes are particularly troubling as modern AI systems are increasingly deployed in high-stakes domains where AI interpretability should be essential for transparency, accountability, and error diagnosis (Mehrabi et al., 2021; Barnes & Hutson, 2024; Ramachandram et al., 2025). Interpretability methods have the potential surface critical failure modes (Kim & Canny, 2017; Zech et al., 2018; Caruana et al., 2015; Meng et al., 2022; Monea et al., 2024; Nguyen et al., 2025) and offer levers for mitigating bias and systematic errors (Arrieta et al., 2019; Kristofik, 2025; Lepori et al., 2025).

Yet, despite frequent analogies to mature sciences like *neuroscience* (Barrett et al., 2019), *biology* (Lindsey et al., 2025), or *physics* (Allen-Zhu & Li, 2023; Allen-Zhu, 2024) of neural networks, the practice of AI interpretability remains in its early foundational stages. Striking dead-salmon artifacts are accompanied by a general statistical fragility: small perturbations to inputs (Ghorbani et al., 2019; Kindermans et al., 2019; Zhang et al., 2025) or changes in random initialization (Adebayo et al., 2018; Zafar et al., 2021) can radically change explanations. Explanations often fail to generalize to new settings and input distributions (Hoelscher-Obermaier et al., 2023). Also, multiple incompatible explanations can be *discovered* for the same behavior (Méloux et al., 2025; Dombrowski et al., 2019). While the dead salmon study demonstrated a simple statistical oversight correctable through multiple comparison adjustments, AI interpretability's difficulties stem from more fundamental issues. In particular, we argue that, for common interpretability queries, computational traces do not uniquely determine explanations.

Beyond neuroscience and AI, such challenges are not unprecedented. Psychology and the social sciences faced a similar reckoning during the replication crisis, when questionable research practices produced widespread false positives (Collaboration, 2015; Simmons et al., 2011; Ioannidis, 2005; Schimmack, 2020). These fields responded with methodological reforms: pre-registration, registered reports, increased statistical power, and explicit multiple-comparison corrections (Munafò et al., 2017; Korbmacher et al., 2023). Likewise, econometrics used causal infer-

ence (Pearl, 2009) to formalize the distinction between correlation and causation, developing identification criteria, sensitivity analyses, and robustness tests (Imbens & Rubin, 2015; Angrist & Pischke, 2009; Heckman, 2007).

Now, AI interpretability can also begin to build its own methodological guardrails. As argued before, this requires both technical innovation and philosophical clarity (Miller, 2019; Williams et al., 2025). This means clarifying our epistemic goals by answering: what does it mean to "explain" a neural network? (Lillicrap & Kording, 2019; Lipton, 2018) Mechanistic interpretability embodies a type of *scientific realism*, aiming to discover the *one true* explanatory algorithm (Psillos, 2005; Chakravartty, 2011). However, there is a significant push-back against the feasibility of this research project (Rudin, 2019; Páez, 2019; Saphra & Wiegreffe, 2024), motivating a shift toward pragmatic approaches prioritizing the utility of the explanations for specific downstream goals (Zou et al., 2025). Here, we align with the pragmatic stance (Dewey, 1948; Chang, 2004; Potochnik, 2017), where explanations are seen as useful models that enable prediction, manipulation, and control (Van Fraassen, 1980; Cartwright, 1983).

**This work.** We analyze failure modes of contemporary AI interpretability methods, ranging from striking dead-salmon false positives to broader forms of statistical fragility, including poor generalization and high variance. We argue that these pathologies share a common root cause: the non-identifiability of many interpretability queries, compounded by the lack of principled uncertainty quantification, where non-identifiability manifests as high-variance estimates that should be reflected in large uncertainty. Diagnosing and addressing these issues, as well as articulating a coherent pragmatic research direction for interpretability, requires reframing AI interpretability as a problem of statistical (causal) inference. Accordingly, we propose one such statistical–causal reframing in which explanations are treated as parameters inferred from computational traces, enabling uncertainty-aware evaluation against meaningful alternative computational hypotheses.

## 2. The Statistical Fragility of AI Interpretability

Reports documenting the failure modes of interpretability methods are frequent and highlight a recurring theme: a general statistical fragility, most strikingly illustrated by dead salmon artifacts. We provide here a non-exhaustive overview of such issues.

**Feature Attribution.** Attribution methods (Simonyan et al., 2014; Sundararajan et al., 2017) aim to highlight input features most relevant to model predictions. However, Adebayo et al. (2018) demonstrated that saliency maps can remain

visually plausible even after model weights are randomized. Further, Dombrowski et al. (2019) showed that gradient-based explanations can be manipulated by adversarial perturbations, leaving predictions unchanged, while Ghorbani et al. (2019) revealed that explanations are unstable under minor data transformations. From a theoretical standpoint, Bilodeau et al. (2024) established impossibility results showing that no attribution method can simultaneously satisfy intuitive desiderata across broad model classes.

**Probing.** Probing methods train a classifier to predict a target label from internal activations. Early studies showed that both linear and structural probes could recover information with surprisingly high accuracy from randomized contextualized embeddings (Conneau et al., 2018; Hewitt & Manning, 2019), and syntactic probes do not generalize (Hall Maudslay & Cotterell, 2021). Later, Ravichander et al. (2021) demonstrated that probes can extract features merely encoded (e.g., inherited from embeddings) even if unused during inference; probing asks whether a concept is encoded in an activation, not whether it is computationally relevant. Capacity-controlled probes (Voita & Titov, 2020; Zhu & Rudzicz, 2020; Pimentel et al., 2020; Belinkov, 2022) or amnesic probing (Elazar et al., 2021) attempt to mitigate such false discoveries.

**Sparse Autoencoders.** Unsupervised concept-discovery pipelines such as sparse autoencoders (SAEs) (Cunningham et al., 2023; Yun et al., 2021; Bricken et al., 2023; Templeton et al., 2024) display analogous pathologies. Heap et al. (2025) showed that SAEs can recover apparently interpretable components even in randomly initialized transformers. Additional studies show that SAEs often fail to generalize across settings or tasks (Heindrich et al., 2025; Kantamneni et al., 2025). Also, Li et al. (2025a) show SAE are sensitive to adversarial input perturbations.

**Concept-Based Explanations.** Concept-based methods (Kim et al., 2018; Bau et al., 2017) aim to identify human-interpretable concepts that align with model representations (e.g., concept activation vectors (Kim et al., 2018) or network dissection (Bau et al., 2017)). These methods also face documented limitations (Sinha & Zhang, 2025; Aysel et al., 2025). Already, Bolukbasi et al. (2021) showed *interpretability illusion* arising where activations of individual neurons in BERT may spuriously appear to encode a concept. Then, Nicolson et al. (2025) showed that concept activation scores can produce inconsistent explanations, and Ramaswamy et al. (2023) show poor generalization and high sensitivity to the dataset used to infer concepts. Finally, Piratla et al. (2024) further demonstrated high variance and recommended incorporating uncertainty estimation.

**Causal Approaches.** To address issues with prediction-based explanations, a shift toward causality-based inter-

pretability has emerged through the use of causal mediation analysis (Pearl, 2012; Elazar et al., 2021; Vig et al., 2020b; Meng et al., 2022; Finlayson et al., 2021; Syed et al., 2023; Monea et al., 2024; Mueller et al., 2024). These methods intervene on intermediate representations to quantify causal effects of components on model outputs. Yet recent work documents substantial fragilities and trade-offs (Canby et al., 2025): Zhang & Nanda (2024) showed that such approaches are sensitive to experimental design. Then, McGrath et al. (2023) discovered a "hydra effect," where ablating components identified as causally important fail to change behavior due to redundant causal pathways. This phenomenon, known as **overdetermination**, occurs when multiple redundant, independently sufficient causal pathways exist (Schaffer, 2003; Sider, 2003; Dyrkolbotn, 2017). Rather than isolating simple mechanisms, interventions tend to reveal overdetermined causal structures.

**Mechanistic Interpretability.** Causal approaches culminate in *mechanistic interpretability* (MI), which aims to reverse-engineer networks into human-interpretable algorithms (Olah et al., 2020). One family of approaches (*where-then-what*) first identifies circuits carrying information from inputs to outputs and then interprets their components (Dunefsky et al., 2024; Davies & Khakzar, 2024; Conmy et al., 2023). The second (*what-then-where*) instead starts from high-level candidate algorithms and searches for causally aligned neural subspaces, using *causal abstraction* metrics (Geiger et al., 2022a;b; Beckers & Halpern, 2019). Despite promising demonstrations, both categories have the typical issues (Sharkey et al., 2025). Subspace patching can produce *interpretability illusions* by activating alternate pathways (Makelov et al., 2023), also a problem of overdetermination. Circuit explanations often fail to generalize (Wang et al., 2022; Li et al., 2025b) and are sensitive to minor experimental choices (Méloux et al., 2025). Exhaustive studies on toy models reveal multiple incompatible explanations for both strategies, even for random networks (Méloux et al., 2025). Finally, Sutter et al. (2025) proved that, in general, existing causal abstraction methods can produce explanations for random networks.

**Natural Language Explanations.** Generating natural language rationales is also a possible approach (Marasovic et al., 2022; Wiegreffe et al., 2022). However, Ajwani et al. (2024) showed that LLM-generated explanations can be systematically unfaithful, confidently providing plausible-sounding justifications for predictions made for entirely different reasons. Moreover, chain-of-thought (self-)explanations are typically unfaithful to the model's computation (Lanham et al., 2023; Arcuschin et al., 2025; Turpin et al., 2023). Since many plausible stories can rationalize any behavior post hoc, natural language explanations are particularly susceptible to confabulation and false positives.

## 3. The Deeper Statistical Issue

The problems documented in Section 2 point to a broad statistical fragility. Here, we identify the common structure underlying these failures: the non-identifiability of interpretability queries.

Behavior-based approaches that study input–output relationships (e.g., feature attributions, behavioral testing) are fundamentally limited by **underspecification**: multiple, distinct explanations can equally well account for the same input–output patterns (Jacovi et al., 2021; Rogers et al., 2021; Hagendorff et al., 2023). Similar observations in cognitive science motivated the development of brain imaging as a complement to purely behavioral data, with the goal of measuring neural computation and thereby obtaining objective, measurable, and more generalizable quantities (Kosslyn, 1999; Logothetis, 2008; Churchland & Sejnowski, 1988). AI interpretability has followed a related trajectory moving toward analyzing internal computation (Mueller et al., 2024). However, predictive approaches based on internal states (probing, SAEs) inherit standard machine-learning pathologies such as overfitting and poor generalization (Belinkov, 2022). These failure modes are instances of **underspecification**: many predictive models can fit the training data equally well, leaving it unclear which ones posit generalizable causal mechanisms (Teney et al., 2022; D'Amour et al., 2022).

Causal approaches, introduced in response to the shortcomings of predictive methods, appear at first to provide the scientific rigor needed for generalizable explanations. However, AI systems are large, distributed systems with many interacting components, which gives rise to redundant and context-dependent causal pathways (Frankle & Carbin, 2019). This creates **overdetermination**, where multiple distinct causal mechanisms are each independently sufficient to produce the same behavior (Tononi et al., 1994; Loosemore, 2012; Sarkar, 2022). Then, finding mechanistic stories within complex computational systems can become *too easy*: many different, incompatible explanations can be produced for the same phenomenon (Lindsay & Bau, 2023; Méloux et al., 2025).

**Identifiability.** These failure modes can be formalized using the concept of identifiability. Informally, identifiability is the property of a statistical inference task stating that the parameters (explanatory variables) of a statistical model can be uniquely recovered from available observations (Casella & Berger, 2024). Identifiability is typically a prerequisite for reliable inference in the natural sciences; without it, inferred explanations remain ambiguous. Therefore, substantial work in statistics, unsupervised learning, and causal inference has focused on characterizing identifiability conditions and designing identifiable tasks (Casella & Berger, 2024; Allman et al., 2009; Locatello et al., 2019; Khemakhem et al., 2020; Shpitser & Pearl, 2008). For interpretability, both underspecification and overdetermination produce non-identifiability, explaining most of the statistical fragilities: (i) **Poor generalization:** when multiple explanations fit the observed data equally well, their explanatory claims can diverge arbitrarily on unseen data. Selecting among these explanations, therefore, depends on arbitrary inductive biases that are rarely validated. (ii) **Sensitivity to design choices:** non-identifiability implies a manifold of explanations that achieve a *good fit*. Different algorithmic choices (datasets, optimization procedures, hyperparameters) traverse this manifold differently, and thus produce different explanations. (iii) **False discovery:** when explanations are non-identifiable, the probability of recovering a spurious explanation that happens to fit the data increases with the size and complexity of the hypothesis space.

Currently, identifiability is just a conceptual analogy, because interpretability has not yet been formalized as an explicit statistical inference task. Making this formal connection and casting interpretability queries as well-specified statistical estimation problems is a necessary first step toward developing methods whose limitations and assumptions can be explicitly characterized.

## 4. The Statistical–Causal Inference Perspective

A straightforward way to address dead-salmon artifacts across interpretability methods is to compare findings on a trained target network against a randomized alternative: the same architecture with randomized weights analyzed by the same method. This leads to a principled hypothesis test against a null hypothesis of randomized computation, an idea foreshadowed in early work on probing (Conneau et al., 2018; Hewitt & Manning, 2019; Ravichander et al., 2021) and circuit discovery (Shi et al., 2024). We formalize such a test in Appendix A and show that, for probing, it eliminates some false discoveries and substantially reduces effect sizes in standard analyses.

While effective, directly correcting dead-salmon artifacts is a very low bar for interpretability. The goal is to address the deeper statistical issues that give rise to these failures in the first place. Nevertheless, hypothesis testing against computationally meaningful null alternatives naturally motivates a broader statistical–causal reframing of interpretability. Here, we sketch one such formalization, viewing interpretability as a problem of *statistical–causal inference*. In this view, explanations are *surrogate models* constructed to answer distributions of causal queries about a computational system. An explanation is useful insofar as it supports prediction and manipulation, generalizes under intervention, and remains robust to noise. This perspective aligns with a growing pragmatist approach to interpretability (Páez, 2019).

## 4.1. Background: Statistical–Causal Inference

Statistical inference provides the rigorous framework through which empirical observations become scientific knowledge (Cox, 2006; Lehmann & Casella, 1998). We argue that interpretability, like every empirical science, must be grounded in these principles. We provide here a brief overview.

**Statistical Models and Identifiability.** A *statistical model* is a family of probability laws $\{\mathbb{P}_\theta^{\mathbf{V}} : \theta \in \Theta\}$ on a sample space $\mathcal{V}$, indexed by parameters $\theta \in \Theta$. Here, $\mathbf{V}$ denotes observed data. Intuitively, we assume data arises from some process indexed by unknown parameters $\theta$, and the goal is to recover $\theta$ from observations. Sound inference requires *identifiability*: distinct parameters must induce distinct distributions over observables. Formally, a model is identifiable if $\theta \neq \theta' \implies \mathbb{P}_\theta^{\mathbf{V}} \neq \mathbb{P}_{\theta'}^{\mathbf{V}}$. Without identifiability, hypotheses cannot be distinguished from data, rendering inference ill-posed.

**Estimators and Uncertainty Quantification.** Given finite observations $\mathcal{D}_n = \{\mathbf{v}^{(i)}\}_{i=1}^n$, an *estimator* $T$ produces an estimate $\hat{\theta} := T(\mathcal{D}_n)$ of unknown parameters $\theta$. Its quality can be assessed through various statistical properties: (i) **Bias**: Does it recover the correct parameter on average? (ii) **Variance**: How much does the estimate vary across datasets? (iii) **Consistency**: Does it converge to the correct parameter as $n \to \infty$? Beyond point estimates, **confidence sets** provide uncertainty quantification under finite sampling.

**Causal Inference.** Many scientific questions go beyond prediction, seeking explanations of *how* variables influence one another. This requires enriching statistical models with a causal structure (Pearl, 2009). Let $\mathbf{V} = \{V_1, \dots, V_d\}$ denote *endogenous variables*, quantities computed within the system. A directed graph $\mathcal{G}$ over nodes $\mathbf{V}$ encodes direct causal relationships: an edge $V_i \to V_j$ indicates that $V_i$ directly causes $V_j$. A *structural causal model* (SCM) is the tuple $\mathfrak{C} = (\mathbf{V}, \mathbf{U}, \mathbf{f}, P_{\mathbf{U}})$, where:

- $\mathbf{U}$ collects *exogenous* (external) inputs representing unobserved causes or environmental randomness, $P_{\mathbf{U}}$ is their joint distribution

- $\mathbf{f} = \{f_1, \dots, f_d\}$ are *structural assignments*, functions that deterministically compute each variable from its causes:

$$V_i = f_i(\mathbf{PA}_i, U_i), \qquad i = 1, \dots, d, \qquad (1)$$

where $\mathbf{PA}_i \subseteq \mathbf{V}$ denotes the parents of $V_i$ in $\mathcal{G}$, and $U_i \in \mathbf{U}$ is its exogenous input.

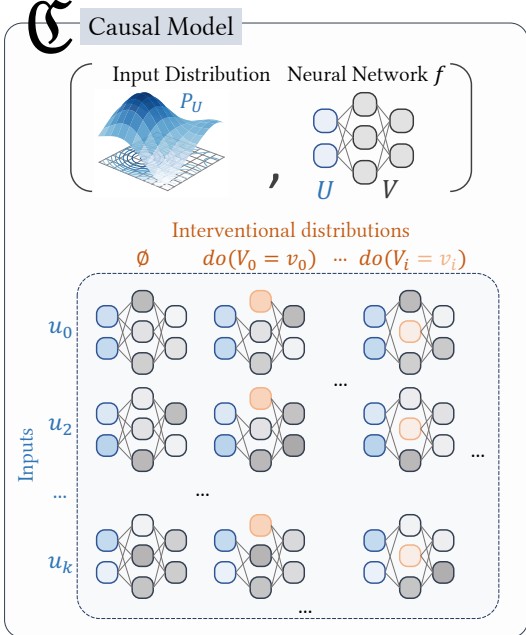

*Figure 2.* The tuple of a target behavior $P_{\mathbf{U}}$ and the computational system with its internal components form an SCM.

SCMs enable reasoning about interventions and counterfactuals. A *hard intervention* $\mathrm{do}(\mathbf{V}_I = \mathbf{v}_I)$ on a subset $\mathbf{V}_I \subseteq \mathbf{V}$ replaces the structural assignments for variables in $\mathbf{V}_I$ with constants $\mathbf{v}_I$, overriding their causal mechanisms. The intervened model $\mathfrak{C}; \mathrm{do}(\mathbf{V}_I = \mathbf{v}_I)$ induces an *interventional distribution* $P_{\mathbf{V}}^{\mathfrak{C};\mathrm{do}(\mathbf{V}_I=\mathbf{v}_I)}$, which captures how the system behaves under this external manipulation.

A *causal query* $q(\mathfrak{C})$ is any well-defined question about the SCM, such as "What is the marginal distribution of $V_i$?" or "What is the average effect of setting $V_i = v$ on outcome $V_m$?" Thus, a causal query is any measurable functional of the SCM, possibly involving conditioning or intervention. Central to causal inference is *query identifiability*: whether $q(\mathfrak{C})$ can be uniquely determined from available observational or interventional data.

## 4.2. Neural Networks as Structural Causal Models

Returning to modern AI interpretability, we first state a standard framing of computational systems as SCMs. Let $f$ be a computational system, typically a neural network, with internal computational elements $\mathbf{V}$ and input distribution $P_{\mathbf{U}}$. The input distribution $P_{\mathbf{U}}$ represents the *behavior of interest* that we aim to explain. For instance, $P_{\mathbf{U}}$ might represent arithmetic prompts to a language model, images from a particular domain, or factual questions about a specific topic.

The tuple $(f, P_{\mathbf{U}})$ naturally defines an SCM $\mathfrak{C} =$

$(\mathcal{G}, \mathbf{V}, \mathbf{U}, \mathbf{f}, P_\mathbf{U})$, where:

- **Endogenous variables V** are the network's computational variables (e.g., hidden states, attention patterns, outputs).

- **Exogenous variables U** are inputs sampled from $P_\mathbf{U}$, representing the behavior we seek to explain.

- **Structural assignments f** are the deterministic functions defining the network's computation (layers, attention mechanisms, nonlinearities).

- **Causal graph $\mathcal{G}$**: the network's computation graph.

The SCM induces a unique *observational distribution* over **V**: sampling corresponds to drawing inputs from $P_\mathbf{U}$, executing a forward pass, and recording desired activations. Also, the SCM encodes *interventional* and *counterfactual* distributions based on external modifications of the inner computation. This perspective is standard within mechanistic interpretability (Olah et al., 2018; Cammarata et al., 2020; Geiger et al., 2022b; 2025) and is illustrated in Figure 2. Then, a *causal query* is any well-specified quantity about $\mathfrak{C} := (f, P_\mathbf{U})$, such as "What distribution would the network produce if we forced activation $V_i$ to value $v$?" or "How much does attention head $V_a$ causally contribute to correct factual recall?"

### 4.3. Explanations as Surrogate Models

In an attempt to provide a general statistical-causal perspective on interpretability, we formalize explanations as *surrogate models*: simpler computational descriptions designed to answer chosen collections of causal queries about a target system. This perspective treats interpretability as a form of model compression, where we seek a simpler model that faithfully approximates a complex system's behavior for queries we care about. In this perspective, every interpretability method is characterized by three ingredients:

1. **Query space $Q$** with **distribution $\mu$**: The set of causal queries to be answered by the explanation. It dictates what aspects of $\mathfrak{C}$ should be explained. This encodes our explanatory goals.

2. **Surrogate set $\mathcal{E}$**: The class of admissible explanations. It dictates what forms the explanation can take, e.g., circuits, sparse subgraphs, linear probes, concept vectors, causal graphs, …

3. **Discrepancy measure $D$**: How we measure whether a surrogate (member of $\mathcal{E}$) *correctly* answers queries.

This framework is (non-rigorously) illustrated with the example of circuit discovery in Figure 3. While standard

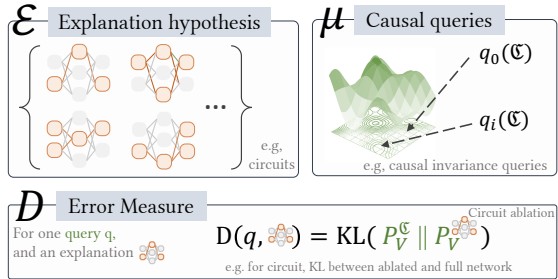

*Figure 3.* An interpretability task is defined by three elements: $\mathcal{E}$, the hypothesis space; $\mu$, the distribution over causal queries about the SCM (model and behavior); and $D$, the error measure.

causal inference often concentrates $\mu$ on a *single* causal query (e.g., average treatment effect), interpretability aims to answer *many diverse queries* drawn from a non-trivial distribution $\mu$. For example, $\mu$ might distribute probability over interventional queries and counterfactual queries across network components, or any functionals of the interventional and counterfactual distributions.

For instance, we can view each candidate explanation $e \in \mathcal{E}$ as defining a *query-answering map* $S_e : Q \to \mathcal{R}$, where $S_e(q)$ is the surrogate's predicted answer to query $q$, and $\mathcal{R}$ is the space of possible answers for query $q$ (e.g., probability distributions, scalar effects, or discrete predictions). The surrogate's *fidelity* is measured by the discrepancy function $D : \mathcal{R} \times \mathcal{R} \to \mathbb{R}_+$ quantifying the error between the answer from $q(\mathfrak{C})$ and the surrogate's prediction $S_e(q)$. Then, we can define the *population risk* of a candidate explanation as the expected error over queries:

$$L_\mu(e) = \mathbb{E}_{q \sim \mu}\big[D\big(q(\mathfrak{C}), S_e(q)\big)\big]. \qquad (2)$$

An ideal explanation $e^* \in \mathcal{E}$ minimizes this risk: $e^* \in \arg\min_{e \in \mathcal{E}} L_\mu(e)$.

**Interpretability Task and Identifiability.** We call the triple $(\mu, \mathcal{E}, D)$ an *interpretability task*, fully specifying what we aim to explain (query distribution $\mu$), what explanations are admissible (hypothesis class $\mathcal{E}$), and how we measure success (discrepancy $D$). The task is *identifiable* if $L_\mu$ admits a unique minimizer in $\mathcal{E}$ (potentially up to predefined acceptable symmetries). Identifiability captures whether the surrogate set can, in principle, be distinguished using the queries deemed relevant by $\mu$. Without identifiability, multiple incompatible explanations achieve the same population risk, making inference fundamentally ambiguous. The analysis of Section 2 indicates that the most common tasks are not identifiable. Finally, this reframing highlights that explanations are *pragmatic computational summaries* of the structure encoded by $\mathfrak{C} := (f, P_\mathbf{U})$. They are inferred models useful for specific explanatory purposes.

**Estimation with Finite Data.** In practice, we face two types of finite sampling difficulties. First, we observe only finitely many queries $q_1, \ldots, q_n \sim \mu$ from the query distribution. Second, for each query $q_j$, we can only collect a finite amount of computational traces by sampling inputs $\mathbf{U} \sim P_{\mathbf{U}}$ and recording the corresponding activations and outputs, potentially under interventions. Let $\mathcal{T}_n$ denote the complete dataset of query-trace pairs. An interpretability method $M$ acts as an *estimator*, mapping this finite dataset to an estimated surrogate explanation:

$$\hat{e} := M(\mathcal{T}_n). \tag{3}$$

A natural estimator is given by empirical risk minimization: choose $\hat{e}$ to minimize the empirical risk $\widehat{L_\mu}(e) = \frac{1}{n} \sum_{j=1}^n \widehat{D}(q_j(\mathfrak{C}), S_e(q_j))$, where $\widehat{D}$ is estimated from finite traces.

Relevant to this exposition, Senetaire et al. (2023) proposed a statistical framing for feature attribution. Then, previous works already explored hypothesis testing and uncertainty quantification for circuit discovery (Shi et al., 2024; Méloux et al., 2025).

### 4.4. Re-Interpreting Documented Issues

The framework does not prescribe which $(\mu, \mathcal{E}, D)$ researchers should adopt. Rather, it provides a *shared language* for making assumptions explicit and rooted in the tools of statistical inference. Different research programs will choose different hypothesis classes or query distributions; the framework ensures that such choices are transparent and their implications are analyzable. Table 1 in the appendix illustrates how existing interpretability methods can be mapped into this formulation, each implicitly making assumptions about queries, surrogates, and error metrics. Under this view, the issues described in Section 2 can be understood as problems of *non-identifiability*.

**Behavioral Benchmarks.** Benchmarks that evaluate model outputs against a gold standard (averaged success over input distributions) ask an identifiable question: *how well does the model perform under a specific task distribution and error metric?* This is arguably the simplest form of interpretability and drove most of the progress in AI. Its usefulness depends on the construction of the benchmark, but the inference problem is well-posed.

**Concept-Based Approaches.** Predictive methods (probes, SAEs) inherit non-identifiability issues from the underlying underspecification of machine learning tasks (D'Amour et al., 2022). For example, methods like Concept Activation Vectors (Kim et al., 2018; Cunningham et al., 2023) postulate that internal states $\mathbf{v}$ are generated by interpretable concepts $\mathbf{z}$ via $\mathbf{v} = g(\mathbf{z})$. This is an instance of (causal)

representation learning, which is **non-identifiable** without auxiliary information (Locatello et al., 2019; Khemakhem et al., 2020). It is therefore unsurprising that proposed improvements mirror standard remedies for underspecification in machine learning: regularization in the form of capacity control for probes (Belinkov, 2022) or cross-validation to assess generalization (Kantamneni et al., 2025).

**Causal Mediation Analysis.** Methods like causal mediation analysis (Meng et al., 2022; Vig et al., 2020a) estimate the indirect effect of a component on observed outputs. As the intervention and model are fully specified, the mediation estimand is unique and **identifiable**. However, the explanatory claim that a component with high effect is the *locus* of a mechanism is **not identifiable**, because of the overdetermined causal structure.

**Circuit Discovery and Causal Approaches.** Circuit discovery seeks a subgraph $G' \subset G$ that preserves the model's performance. This task faces the "Hydra effect" (McGrath et al., 2023) and causal overdetermination. If parallel pathways $A$ and $B$ are sufficient, circuits containing only $A$ or only $B$ both satisfy fidelity criteria. Thus, even *correct* causal methods may recover many different explanations consistent with the same behavior (Méloux et al., 2025). Addressing this requires formulating identifiable causal questions. Causal abstraction (Beckers & Halpern, 2019; Geiger et al., 2025) offers a promising direction, as it operates at a coarser representational level where overdetermination can be absorbed into the abstracted representations. However, current operational metrics demonstrate empirical non-identifiability (Méloux et al., 2025; Sutter et al., 2025).

## 5. Discussion

The systematic failures documented in Section 2 demanded an explanation. We have argued that these pathologies share a common root cause: **non-identifiability**. Most current interpretability tasks attempt to infer explanatory structures that are not uniquely determined by available computational traces. To trace a path forward, we proposed one formalization of interpretability as statistical-causal inference. This framework is tentative rather than definitive; we encourage the community to improve upon it. The important aspect is the *methodological commitment* to making assumptions explicit and quantifying uncertainty rigorously.

### 5.1. Advantages of the Statistical-Causal Perspective

Drawing on the philosophy of science (Chang, 2004; Woodward, 2003; Potochnik, 2017) and recent calls for a pragmatic approach to interpretability (Davies & Khakzar, 2024; Williams et al., 2025), the framework naturally distinguishes the *explanandum* (what is to be explained, encoded in $\mu$)

from the *explanans* (what does the explaining, encoded in $\mathcal{E}$). Researchers and practitioners have substantial freedom in choosing both. There is no single "correct" explanation of a neural network. The appropriate type of description depends on one's purposes (Potochnik, 2017). Descriptive understanding corresponds to queries about observational distributions; predictive goals involve queries requiring surrogates to generalize to new input distributions; control and intervention require queries about counterfactual or interventional distributions.

However, once the explanatory project is specified, i.e., once $(\mu, \mathcal{E}, D)$ are fixed, the explanation becomes an **objective inference problem**. The best surrogate $e^* \in \mathcal{E}$ is the one minimizing $L_\mu(e)$, and is a property of the system itself and the interpretability task. If the task is identifiable, this explanation is unique (up to permissible symmetries, e.g., rotation invariance in representation space). This reconciles pluralism about explanatory goals with rigor about explanatory claims.

Perhaps most critically, the statistical framing demands that interpretability methods report not just point estimates but *confidence sets* or *posterior distributions* over explanations (in case of Bayesian framing). Just as we would not trust a clinical trial reporting effect sizes without confidence intervals, we may not trust interpretability claims without uncertainty quantification. When explanations are non-identifiable, this uncertainty will be large; when they are identifiable with finite data, uncertainty shrinks as observations accumulate.

## 5.2. Towards Useful and Identifiable Interpretability Tasks

Identifiability is not an intrinsic property of the model under study but of the interaction between $\mu, \mathcal{E}, D$, the model $f_{NN}$, and the behavior of interest $P_{\mathbf{U}}$. We might wonder what choices to make in order to improve the identifiability and usefulness of interpretability queries.

**Query richness.** The queries in the support of $\mu$ must be sufficiently discriminative to distinguish candidate explanations in $\mathcal{E}$. There is a fundamental trade-off between discriminative power and sample efficiency. If $\mu$ spreads probability mass over a large support, accurately estimating $L_\mu$ may require prohibitive amounts of interventional data. Conversely, concentrating $\mu$ on too few queries risks not singling out one explanation in $\mathcal{E}$.

**Expressivity vs. parsimony in $\mathcal{E}$.** Conversely, the hypothesis class must have sufficient capacity to approximate the queries well (low bias) but not so much flexibility that many distinct explanations all achieve low error (large equivalence classes, high variance, non-identifiability). This is akin to the classical bias-variance tradeoff, pointing toward standard fixes like regularization of the hypothesis class (Belinkov, 2022).

**Human cognitive constraints.** Interpretability is meant to facilitate *human understanding*. Empirical studies suggest people can mentally simulate models with only a handful of interacting components (Lombrozo, 2006; Wilkenfeld, 2013; Keil, 2006; Hassija et al., 2024). Explanations exceeding these structural limits may be technically *correct* yet fail to provide insight. Designing $\mathcal{E}$ with human simulability in mind ensures that understanding remains the end goal.

## 5.3. Opportunities for Future Work

**Characterizing identifiability conditions.** A systematic theoretical program could characterize when specific $(\mu, \mathcal{E}, D)$ triplets are identifiable, mirroring similar efforts in causal inference (Shpitser & Pearl, 2008) and unsupervised learning (Locatello et al., 2019; Khemakhem et al., 2020). What symmetries and invariances are unavoidable in representation space, and when is identifiability up to such equivalences acceptable? Constructing a taxonomy of identifiable interpretability tasks would provide actionable guidance for practical scenarios.

**Bayesian interpretability and uncertainty quantification.** Bayesian approaches offer an elegant framework for handling non-identifiability and quantifying uncertainty (Gelman et al., 2013). Specifically, one could specify a **prior distribution** $\pi(e)$ over the explanation class $\mathcal{E}$, encoding structural preferences (e.g., sparsity, modularity) or incorporating prior information from related studies. Then, the **likelihood model** $P(\mathcal{T}_n \mid e)$ describes how computational traces are generated given explanation $e$. Finally, the **posterior updates** via Bayes' rule: $\pi(e \mid \mathcal{T}_n) \propto P(\mathcal{T}_n \mid e)\pi(e)$, refines beliefs as observations accumulate. Then, **credible sets** can quantify uncertainty. When explanations are non-identifiable, the posterior remains diffuse across an equivalence class; uncertainty quantification naturally reflects this fundamental ambiguity. Conversely, as more discriminative queries are observed, the posterior concentrates. This further provides a principled framework for active setup: strategically selecting queries from $\mu$ that maximally reduce posterior uncertainty.

**Meta-analysis and cumulative science.** Meta-analytic methods (Borenstein et al., 2021) could coherently aggregate evidence across studies, accounting for heterogeneity in $\mu, \mathcal{E}$, and experimental conditions. Standardized effect size measures, pre-registration of analyses, and open sharing of collected computational traces would enable interpretability to become a cumulative science where knowledge systematically builds over time. In general, responses proposed by other fields (Poldrack et al., 2017; Korbmacher et al., 2023) become available for interpretability.

## Impact Statement

This work aims to improve the scientific rigor and reliability of Mechanistic Interpretability (MI). As MI techniques are increasingly proposed for safety auditing, model alignment, and regulatory compliance, it is critical that these methods produce stable and statistically valid explanations. Our research highlights the risks of relying on unstable point-estimates, which can lead to unjustified confidence in a model's safety properties or internal mechanisms. By advocating for statistical robustness and best practices in circuit discovery, this work contributes to the development of more trustworthy AI systems and helps ensure that future interpretability tools provide a solid foundation for policy and safety decisions.

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

# A. Fixing Dead Salmons with Hypothesis Testing

Consider an interpretability method $M$ that aims to explain a neural network $f$ for some input behavior $P_U$, we note $\mathfrak{C}$ the tuple $(f, P_U)$ as done in the main paper. The method produces an explanation $\hat{e}$ from finite observations from $\mathfrak{C}$, possibly under interventions. This tentative explanation could take the form of a circuit, a set of important features, concept activation vectors, or any other hypothesis class. We might wonder how to prevent dead salmon artifacts from arising with the interpretability method $M$?

A simple, direct, and natural solution is to frame this question as a hypothesis test against a null hypothesis where the observed explanation arises from random computation. Already, for probing, Ravichander et al. (2021) discusses the possibility of comparing the probe against a probe trained on random embeddings. The general idea is to construct a family of null models represented by a distribution $P_{\tilde{\mathfrak{C}}}$, which preserves the network's architectural properties while disrupting the specific computational mechanisms we aim to explain. Such null models can be obtained, for example, via full weight randomization, random orthogonal transformations of representations, or label shuffling (recovering the standard permutation test).

For a given interpretability method, we define a test statistic $T(\hat{e}, \mathfrak{C})$ that quantifies explanatory fit for the interpretability task at hand. For example, for probing methods, $T$ could be test accuracy; for circuit discovery, $T$ could measure behavioral fidelity; for attribution methods, $T$ could quantify the correlation between attribution scores and actual intervention effects.

Applying the intepretability method $M$ to one null model $\tilde{\mathfrak{C}}^{(b)}$ from the randomized family yields explanations $\tilde{e}^{(b)} = M(\tilde{\mathfrak{C}}^{(b)})$ and corresponding null statistics $T_{\text{null}}^{(b)} = T(\tilde{e}^{(b)}, \tilde{\mathfrak{C}}^{(b)})$. Then, following standard procedure, the Monte Carlo estimated $p$-value is:

$$\hat{p} = \frac{1 + \sum_{b=1}^{B} \mathbb{I}\{T_{\text{null}}^{(b)} \geq T_{\text{obs}}\}}{B + 1}, \qquad (4)$$

where $T_{\text{obs}} = T(\hat{e}, \mathfrak{C})$. The addition of 1 to both the numerator and denominator ensures Type I error control: $\Pr(\hat{p} \leq \alpha \mid H_0) \leq \alpha$, where $H_0$ is the null hypothesis (North et al., 2002; Phipson & Smyth, 2010). By design, when the randomization includes full weight reinitialization, no dead salmon artifacts can remain.

## A.1. Experiments

To illustrate the hypothesis test, we experiment with three probing tasks.

**Sentiment Analysis (IMDb).** We reuse the IMDB sentiment classification setup from Figure 1. For each layer of `BERT-base-uncased`, we extract the average sentence embedding and train a linear probe to predict binary sentiment. We also train probes on $k=20$ random reinitializations of the model, and evaluate statistical significance using the hypothesis test described above. All probes are trained and evaluated on 1000 sentences with 10-fold cross-validation. Figure 4(A) reports (i) the average probe accuracy at each layer for the pretrained model, the randomized models, and a random guessing baseline, and (ii) the corresponding effect sizes relative to random guessing and to randomized models. While all pretrained layers outperform random guessing with large effect sizes, none are statistically distinguishable from the random reinitializations under the new test. Later layers, however, show a clear upward trend in effect size relative to randomized models.

**Syntactic Structure (POS Tagging).** We next assess token-level syntactic information using POS-tagging probes (Tenney et al., 2019). For each layer of `BERT-base-uncased`, we extract contextual token embeddings and train logistic regression probes on a subset of CoNLL-2003, one probe per layer that should work for all tokens and all POS tags. As above, we also train probes on $k=20$ random reinitializations and apply the same statistical test. Probes are evaluated with 10-fold cross-validation on 500 sentences. Figure 4(B) reports the layer-wise probe accuracy and effect sizes relative to a majority baseline and to randomized models. Consistent with prior work, POS accuracy peaks in middle layers (Tenney et al., 2019). However, when tested against randomized models rather than random guessing, only the middle layers remain statistically above chance, and the effect sizes are substantially reduced. This shows that testing against random computations eliminates many positive findings while still allowing for genuine positive discoveries where structure is robust.

**World Models (Space and Time).** Finally, we investigate the emergence of linear representations of space using the "world places" dataset from (Gurnee & Tegmark, 2024). Using `pythia-160m`, we extract average residual stream activations on each token of place names and train linear ridge regression probes to predict their geospatial coordinates (latitude and longitude). We compare the pretrained model against $k = 20$ baselines where transformer block weights are randomized while embeddings remain fixed. Probes are evaluated using $R^2$ scores with 10-fold cross-validation. Figure 4(C) reveals that raw embeddings (Layer 0) contain latent spatial structure ($R^2 \approx 0.12$), significantly outperforming random guessing ($Z \approx 100$). Passing these embeddings through randomized transformer blocks decreases linear readout ($R^2 \approx 0.38$). In contrast, the pretrained model's layers slightly improve this spatial linearity relative to the random baseline, suggesting that deeper layers progressively construct a more coherent spatial representation.

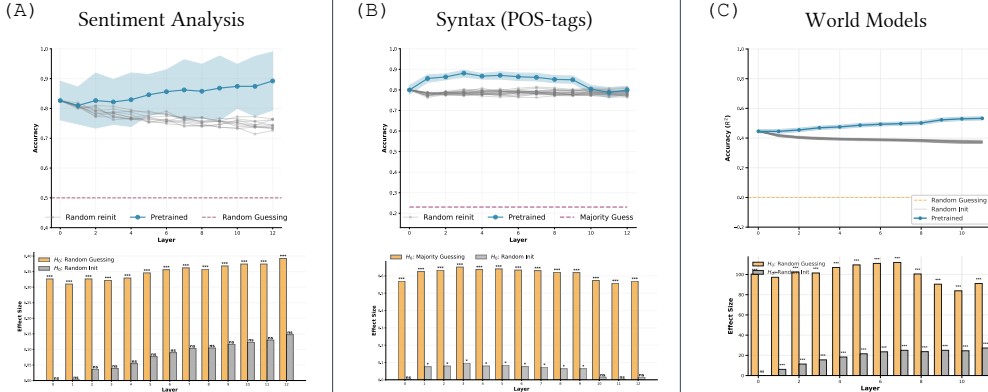

*Figure 4.* (A) Sentiment analysis experiment where probes on pretrained BERT are compared against probes trained on random computation. (B) Same experiment based on predicting syntactic labels (POS tags). (C) Reproducing the first experiment of Table 2 in (Gurnee & Tegmark, 2024), probing for indications of world models on pythia-160m.

By the final layers, the learned structure statistically surpasses the random baseline ($Z \approx 25$), confirming that the model eventually learns to encode space explicitly beyond the geometry inherent in the embeddings.

| Method | Hypothesis space $\mathcal{E}$ (surrogate model) | Typical causal query $q(\mathfrak{C})$ and error criterion $D$ |
|---|---|---|
| **Performance benchmarking** | Single scalar summarizing predictive performance (e.g., accuracy, calibration, perplexity). | *Observational query:* output score distribution under $P_{\mathbf{U}}$. Error: difference in expected performance metrics, e.g., $\lvert\mathbb{E}[f(\mathbf{U})] - \mathbb{E}[\hat{f}(\mathbf{U})]\rvert$. |
| **Probing (linear / diagnostic classifiers)** | Linear or shallow classifiers mapping internal activations to target variables (e.g., part-of-speech tags). | *Observational query:* conditional distribution $P(Y \mid \text{activations})$. Error: classification loss. |
| **Feature attributions (saliency, SHAP, Integrated Gradients)** | Input-level additive surrogates assigning contribution scores so that $f(x)$ is approximated by $\sum_i e_i(x)\, x_i$ relative to a baseline. | *Counterfactual queries:* local (additive) approximation of model behavior around inputs $x$. Error: fidelity loss between model predictions and surrogate reconstruction |
| **Concept-based methods (e.g., TCAV, ACE, concept bottleneck models)** | Surrogates mapping internal activations to interpretable concept variables and modeling $f$'s dependence on them. | *Interventional queries:* model sensitivity or dependence on interpretable concept activations within latent space. Error: deviation between surrogate-predicted and model-predicted sensitivities (e.g., directional derivative mismatch). |
| **Circuit discovery** | Subgraphs of the computational graph representing causal mechanisms. | *Interventional queries:* outputs distribution under targeted ablations encoded by the circuit. Error: consistency in output distribution, e.g., $KL\big(P_{\mathfrak{C}}(Y \mid U) \,\Vert\, P_{\text{circuit}}(Y \mid U)\big)$ |
| **Causal tracing (patching, mediation analysis)** | Scalar importance scores over units or connections inferred from intervention or mediation effects. | *Counterfactual mediation queries:* total, direct, or indirect effect of node $V_i$ on target $Y$. Error: difference between predicted and empirical effects. |
| **Causal abstraction / model-level alignment** | High-level structural causal model with mappings from low-level network variables $\mathbf{V}$ to abstract variables $\mathbf{Z}$. | *Interventional invariance queries:* the actions of abstracting from $V$ to $Z$ and intervening should commute. Error: causal abstraction error, measuring causal alignment as violation of commutative properties of abstraction and intervention. |

*Table 1.* Interpretability methods as instances of the statistical–causal framework of *surrogate models*. Each method specifies a hypothesis class $\mathcal{E}$, causal query family $q(\mathfrak{C})$, and associated error measure $D$ quantifying how faithfully the surrogate answers the queries.

