# OpenReview forum: "The Dead Salmons of AI Interpretability"
_ICML.cc/2026/Conference — Submitted to ICML 2026_

### Official Review · Reviewer_QrAu · 2026-03-09

**Soundness:** 2
**Presentation:** 3
**Significance:** 2
**Originality:** 3
**Overall Recommendation:** 3
**Confidence:** 3

**Summary:**

This paper analyzes failure modes of contemporary XAI methods, from Feature Attribution to Mechanistic Interpretability. It uses the dead-salmon metaphor to explain the statistical fragility: (i) Poor generalization, (ii) Sensitivity to
design choices, (iii) False discovery. The authors further argue that these issues share a common root cause: the nonidentifiability of many interpretability queries. With this goal in mind, the authors propose a statistical–causal reframing in which explanations are treated as parameters inferred from computational traces, enabling uncertainty-aware evaluation against meaningful alternative computational hypotheses.

**Compliance With Llm Reviewing Policy:**

Affirmed.

**Final Justification:**

I've updated my assessment to a 3 (Weak Reject) after reading through the discussion, especially CTuq's argument regarding venue fit. I still think the core idea of bridging causality, statistical inference, and XAI is novel and interesting. However, I have to agree that the paper reads more like a position paper than a standard ICML main-track submission. The empirical contribution is too thin to carry the main text's theoretical weight, and burying it in the appendix is an unorthodox choice. Without strong experiments, I am not fully convinced by the presented framework itself, so I highly encourage the authors to strengthen their empirical validation in future revisions. Because it focuses heavily on conceptual framing at the expense of practical execution, I am lowering my recommendation to a 3 as a main track paper. That said, I am fully supportive of whatever final call the AC makes.

**Key Questions For Authors:**

My biggest concern with this paper is how your theoretical framework can be applied practically. Could you specifically address the practical implementation challenges raised in W1: how can our explanatory goals be written down into a mathematically precise probability distribution?

**Limitations:**

yes

**Strengths And Weaknesses:**

**Strengths**:

1.  The paper uses an effective analogy from neuroscience, such as the "dead salmon" study, to explain the false positive issues in current XAI research.

2. The authors analyze in detail the failure modes of contemporary XAI methods, ranging from feature attribution to mechanistic interpretability, and attribute these causes to deeper statistical insights: underspecification in explanations and overdetermination in model reasoning. The argument looks solid.

3. The paper proposes a rigorous hypothesis-testing approach based on statistical inference against the null hypothesis of randomized computation, providing a rigorous way to reject spurious explanations. The framework also seems to handle different form of explanations.

---

**Weaknesses**:

1. While the proposed $(\mu, \mathcal{E}, D)$ framework is conceptually appealing, the paper lacks a clear roadmap for how this theory can be applied to real-world XAI practice. For instance, in Sec 4.3, the paper states that every interpretability task requires a Query space with distribution $\mu$, and this distribution encodes our explanatory goals; they do not provide a concrete method for how the explanatory goals can be mathematically formalized.

2. The empirical results are deferred to the Appendix and are largely restricted to small-scale experiments such as Sentiment analysis on BERT. In addition, the experiments are repeated on 20 random reinitializations, which raise somecerns on statistical significance given BERT can have a massive hypothesis space (of explanations). Also, how does your framework generalize to SOTA architectures like  LLMs or Vision Transformers?


3.  The non-identifiability argument suggests that many current explanations/queries are fundamentally ill-posed. However, the paper does not offer a solution to fix them. This makes the findings and arguments feel more like an academic critique than a practical engineering solution.

---

> ### Author Rebuttal · Authors · 2026-03-27
>
> We thank the reviewer for the positive views on our paper and the thoughtful comments.
>
> ## Practical roadmap
> Our framework does provide a concrete and actionable methodology for approaching interpretability problems. We now include a more explicit statement in the discussion section. The workflow is as follows:
> - (1) Formalize the explanatory goal via a choice of query and error metric (see below).
> - (2) Analyze identifiability: determine whether the explanation is identifiable. This step maps uses the framework to map to well-established tasks from statistical and causal inference.
> - (3) Construct an estimator: if identifiability holds, estimate the target along uncertainty. If some non-identifiability remains, characterize the equivalence class of explanations (e.g., the set of explanatory circuits in MI instead of just one)
> - (4) Quantify uncertainty and variability in the inferred explanations to estimate the quality of the findings.
> - (5) Validate assumptions via hypothesis testing (as described in Appendix A), including tests against randomized or alternative computational hypotheses.
> - (6) Report results in a form that makes explicit the assumptions made in (1) and (2) and the remaining uncertainty from step (4).
> This pipeline turns interpretability into a standard inferential problem and enables cumulative progress (e.g., comparison across studies, meta-analysis).
>
> This pipeline clearly outlines a roadmap for future research directions: how to best formalize the various types of explanatory goals we might have (step 1), how to study identifiability, especially with respect to causal overdetermination which has not been studied a lot in the field of causality yet (step 2). How to craft computationally efficient uncertainty-aware estimators (step 3).
>
> Following the reviewer’s suggestion, we have restructured the discussion section to highlight this practical roadmap.
>
> ## Formalizing explanatory goals
> In our framework, explanatory goals are made precise through queries and error measures. A choice of queries (and the distribution) specifies what property of the computation an explanation should capture. This is a choice made by researchers to model their goals and it is subject to criticism by others. For example, in circuit discovery, the queries are invariances of the input-output behavior under interventions (ablations) on the computational graph. The distribution over queries is how researchers sample the interventions which has not been made explicit until now by circuit discovery research but was implicitly the cause of debate (e.g., the debate on "faithfulness" via sampling interventions outside of the candidate circuit in Hanna et al. (COLM 2024). Table 1 already provides explicit formulations for several common methods.
>
> ## Academic critique vs practical contribution
> While our first contribution is a diagnosis of failure modes, this diagnosis is not purely conceptual, it is also operational.
> By grounding the analysis of these failures in the well-known fields of statistical and causal inference using the precise concept of non-identifiability, the framework:
> - Explains a wide range of empirical pathologies (e.g., sensitivity, poor generalization, false discoveries).
> - Provides concrete tools, notably hypothesis testing against computational alternatives, which eliminate “dead salmon” effects across methods.
> - Provides a pipeline (described above) and concrete motivation and open problems for future work to study (identifiability because of causal overdetermination)
> We have more details on practical examples in our answer to reviewer wdR8.
>
> ## How to address non-identifiability in practice
> We agree that it remains a big challenge. Our framework provides a general strategy by making the bridge between XAI settings and statistical--causal inference settings where identifiability has been studied for a long time. In some instances, we can already use the bridge to decide clear non-identifiability problems (see Section 4.2 and answer to reviewer MJG4 about gradient-based feature attributions). Some other cases like causal overdetermination are more complicated and less studied but our work provides a strong motivation for future theoretical work on these topics. Finally, in our answer to reviewer MJG4, we detail the addition of a limitations section that will include this point.
>
> ## How does the framework generalize to other architectures?
> The hypothesis testing framework is architecture-agnostic and even generalizes across interpretability settings.
> We repeated the experiments in Figure 1 and Figure 4 (appendix) with 2 more modern models: modernBERT-base as a modern encoder model and Qwen3-1.7B as modern decoder and use 100 random repeats instead of 20. The findings remain consistent (dead salmon emerging Figure 1 and 4 (A) and being fixed by the hypothesis testing framework), reinforcing our claims. Large random repeats only decrease the confidence intervals but do not change the findings.

---

> > ### Author Rebuttal · Reviewer_QrAu · 2026-04-01
> >
> > The roadmap makes sense to me. However, I still find that the proposed framework could be challenging to apply in real-world settings due to issues of non-identifiability and scalability. That said, from a purely theoretical perspective, I believe this is a novel piece of work, and I will therefore maintain my positive assessment.

---

> > > ### Author Response · Authors · 2026-04-07
> > >
> > > Thank you again for your thoughtful engagement and positive assessment.
> > >
> > > We agree that non-identifiability and scalability can make the framework challenging to apply in practice. Importantly, non-identifiability is precisely one of the core challenges our work aims to highlight. The framework aims to help distinguish between settings where better methods can resolve issues and those where interpretability questions themselves may need to be reformulated. In that sense, we see the identification of the non-identifiability challenge as one central contribution rather than a limitation.
> > >
> > > We are grateful for your recognition of the novelty of this perspective and for your constructive feedback throughout the process, which really helped improve the revision of this work.

---

### Official Review · Reviewer_wdR8 · 2026-03-11

**Soundness:** 4
**Presentation:** 4
**Significance:** 2
**Originality:** 3
**Overall Recommendation:** 5
**Confidence:** 4

**Summary:**

This paper focuses on some of the issues and limitations of interpretability and suggests these issues, particuarly the problem of identifiability, may be alleviated  through a causal-statistical and pragmatic reframing of interpretability. The paper is therefore mostly making a conceptual contribution by first identifying and synthesising literature on issues with interpretability and secondly by arguing why a reframing could alleviate these issues. The suggested framing identifies an 'interpretability task' which is a tuple containing a query space, a surragote set of admissible explanations and a discrepancy measure of how well an admissible explanation answers a query in the query space.

**Compliance With Llm Reviewing Policy:**

Affirmed.

**Final Justification:**

My final recommendation is accept. After reading the author(s) rebuttal I increased my score from weak accept to accept as they addressed the  questions and comments I made, some also addressed in other review rebuttals. I think the paper makes an original contribution to pragamtic XAI and the presentation is very good.

**Key Questions For Authors:**

Could you include more detail on how a user could construct their query space, surrogate set and discrepancy measure, e.g. through a worked example? I would like to see how this framing could help select a method for a predefined task, e.g. does the framing help decide between using a causal or non-causal method, or something more specific like SHAP over PDP? Table 1 is slightly more general than this.

**Limitations:**

Discussion of limitations is weaved in but for clarity it could be useful to have a stand out limitations section in the discussion.

**Strengths And Weaknesses:**

The paper is well written, organized clearly and mostly well placed in the literature, although I think would benefit from some additional comparisons to similar pragmatic and/or statistical-causal reframing suggestions such as Bordt et al.'s 2025 position paper (Position: Rethinking Explainable Machine Learning as Applied Statistics) or Sullivan's SIDEs framework (2024). The core contribution comes from the interpretability task definition (sections 4.3 and 4.4), with the majority of the paper being background and positioning work. Sections 4.3 and 4.4 are quite short as the core contribution and I think could be extended to include something like a worked example to see how someone could use this new framing in practice. I think it could also benefit from an explanation of how to select the tuple; how does a user select the 'admissible explanations' for a particular query space? Table 1 helps with this for a technical audience (can you move it so it isn't at the end of the appendix) but it might be difficult for someone non-technical to use and therefore could run into HCI complications. I like the framing but I think it requires a bit more to make it concrete and applicable.  Although I think it is slightly lacking in significance because of this, I think it would be a worthwhile contribution to pragmatic accounts of XAI/interpretability and therefore I lean towards accept. (I think it would make a strong position paper).

---

> ### Author Rebuttal · Authors · 2026-03-27
>
> We thank the reviewer for their positive views on our work and their insightful and constructive feedback.
>
> ## Adding discussion to Bordt et al. and Sullivan
> We agree and have included both works in the introduction in the same paragraph where we discuss the work by Senetaire et al and Shi et al (see our answer to reviewer MJG4). Sullivan's work also serves as a good motivation for the pragmatic approach to XAI and our framework provides a compatible implementation of the idealization perspective via surrogate models. Similarly, Bordt uses a statistical lens (the statistics of functions) which is fully compatible with our framework. Additionally, our work has a diagnosis perspective on repeated failures of most XAI methods: we identify a common failure mode across approaches (the tendency to produce false positives) and analyze it through statistical and causal concepts, especially identifiability. We believe this unifying perspective, grounded in documented failure cases in sections 2 and 3, is itself novel and forms an important contribution of our work.
>
> ## Practical usage of the framework and worked example:
> We agree that making the framework concrete is essential. First, we want to emphasize that the hypothesis testing proposed in the paper is one simple and direct outcome of the framework that does yield very concrete applications by removing all types of dead salmons in all interpretability at the cost of additional computation.
>
> For example, when adopting our framework, probing is naturally cast as hypothesis testing which forces us to ask: what are the meaningful alternative hypotheses? For computational models, these would be computational alternatives: models trained on other datasets, other tasks, partially trained or not trained. This naturally lead to the hypothesis testing framework we outline in the appendix, which provides can discover and remove dead salmon effect but also produced revised effect sizes of previous works (Figure 4). Instead of current methods looking for evidence of success, the hypothesis testing framework shifts the focus to falsification which eliminates a lot of issues. Based on your suggestion, we now reframe the story of hypothesis testing for probing that we already present in the paper as a more detailed worked example in an appendix B that accompanies Table 1.
>
> Second, subtleties of causal inference become clear with the framework. For example, in causal mediation analysis (CMA) once formalized (as done in Table 1 at the high-level) it becomes clear that CMA is indeed identifiable but only at the sample level (one input), aggregating CMA scores over many inputs (population-level) is not well-justified. This subtlety, well known in causal mediation analysis, becomes clear because CMA is based on a set of counterfactual queries. This should prevent over-interpretation of aggregated CMA scores.
>
> Finally, note that in our answer to reviewer MJG4, we discuss the addition of a short paragraph summary in section 4.2 explaining how non-identifiability emerges in XAI from non-identifiability in standard tasks. Also, in our answer to reviewer QrAu, we discuss how we restructure the discussion with a concrete pipeline to approach an interpretability task using our framework (formalize goals, check identifiability and potential assumptions needed, craft estimators with proper uncertainty quantification, find best fitted explanation, test hypotheses in particular test whether assumptions hold -> others can then critique the formalization and the assumptions). We believe that these small reframings focusing on practical recommendations and practical usage should improve the concrete applicability of the proposed framework.

---

> > ### Author Rebuttal · Reviewer_wdR8 · 2026-04-01
> >
> > My questions are addressed, I will increase my score from weak accept to accept.

---

### Official Review · Reviewer_MJG4 · 2026-03-12

**Soundness:** 3
**Presentation:** 2
**Significance:** 3
**Originality:** 3
**Overall Recommendation:** 5
**Confidence:** 3

**Summary:**

The paper addresses the failure of XAI methods from a causality perspective, aiming to create statistical-causal framework that can be used to diagnose the source of errors. It frames neural networks as SCMs and maps different problems of XAI to non-identifiability of the parameters of the models.

**Compliance With Llm Reviewing Policy:**

Affirmed.

**Final Justification:**

I have increased my score, as I believe that, given the changes that the authors promise, this paper is a valuable contribution to both the XAI and the causality community.

**Key Questions For Authors:**

See weaknesses

**Limitations:**

The authors do not explicitly discuss limitations, and I believe that the paper would profit from a discussion on the limitations of this framework.

**Strengths And Weaknesses:**

Strengths:
* Well-organized synthesis of failures across methods
* The identifiability framing is novel and unifying
* This paper gives the community a shared language that makes assumptions explicit and comparable

Weaknesses:
* The relationship to prior work (Senetaire et al., Shi et al.) needs sharper delimitation. What exactly is new here? This would strengthen the contribution.
* Section 4.4 skips explanations for some of the failure modes. For example, saliency methods like LRP fail the randomization sanity check because backpropagation converges toward a rank-1 structure (see Sixt et. al, 2019). How does this failure mode fit (or not) into this framework?
* I think the paper still lacks a clearer description of how non-identifiability manifests across the different methods, in order to make the paper accessible also to readers which do not come from the causal community.

---

> ### Author Rebuttal · Authors · 2026-03-27
>
> We thank the reviewer for their positive assessment and helpful suggestions, which have helped improve the paper.
>
> ## Clarifying the relationship to some prior works.
> We agree that our positioning relative to the prior work you mentioned can be emphasized. Works such as Senetaire et al. and Shi et al. use statistical language to analyze specific interpretability settings: feature attribution and circuit analysis, respectively. Our contribution differs across several axes:
>
> (i) broader scope: we provide a unified treatment across multiple interpretability paradigms (e.g., attribution, probing, mechanistic interpretability) beyond a specific method per setting,
>
> (ii) Unified technical diagnosis: we identify a common failure mode across settings (the tendency to produce false positives) and discuss it via the formal lens of statistical and causal concepts, in particular identifiability.
>
> (iii) General framework: we introduce a statistical–causal framework that subsumes these prior approaches as special cases, enabling a consistent analysis across methods.
>
> We have revised the introduction to make this distinction explicit with two other motivating works mentioned in answer to reviewer  wdR8) and mention in section 4 that these prior works instantiate special cases within our framework.
>
> ## Non-identifiability interpretation of LRP failures (Sixt et al.)
> Thank you for pointing out this important connection that helped us improve section 4.2 with a short paragraph dedicated to feature attribution. Incorporating it gives another concrete example of how non-identifiability arises, this time for gradient-based feature attribution methods. Indeed, the phenomenon observed by Sixt et al. nicely fits our narrative and can be understood as follows: The rank-1 collapse when moving backward to earlier layers means the mapping from model parameters to explanations is many-to-one, i.e., different parameterizations yield almost identical saliency maps. This implies that explanations are not identifiable from the computation (at later layers). Sixt et al. provide an explanation for how this non-identifiability emerges in gradient-based feature attribution approaches.
>
> ## Clarifying how non-identifiability manifests
> We agree that this can be made more accessible. To address this, we have added a high-level summary paragraph at the end of section 4.2 with the following intuition:
> Interpretability methods can be viewed as (explicitly or implicitly) solving proxy statistical inference problems. These problems may suffer from non-identifiability, which, if unaccounted for, leads to high false positive rates. Different classes of methods inherit different forms of non-identifiability depending on the underlying statistical task:
> - Probing → supervised learning → non-identifiable predictors without appropriate assumptions or regularization
> - Concept-based methods → latent variable models → non-identifiability without structural constraints
> - Attribution, circuit analysis, causal abstraction → causal inference / causal representation learning → non-identifiability under insufficient interventions and empirical causal overdetermination.
>
> This addition makes explicit how standard statistical non-identifiability translates into observed XAI failures, and how our framework makes the bridge between interpretability and statistical inference to help discuss and address these issues.
>
> ## Limitations section
> We appreciate the suggestion to include a dedicated limitations section. We have added one covering the following points:
> - The framework still requires specifying the query distribution which remains a difficult question in general (see Discussion section) but a necessary one to engage with.
> - It could be that interesting interpretability questions may be fundamentally non-identifiable. Although this risk reinforces, rather than diminishes, the need for explicit reasoning about identifiability.
> - Uncertainty quantification (e.g., hypothesis testing) may require re-running interpretability pipelines under alternative hypotheses, which can be computationally expensive.
> - The paper does not provide a complete solution to non-identifiability, but it establishes a bridge to statistical methodology, where these issues have been extensively studied. This enables the import of principled tools, as illustrated by our example based on hypothesis testing (e.g., addressing "dead salmon"-type effects).

---

> > ### Author Rebuttal · Reviewer_MJG4 · 2026-04-01
> >
> > The authors address all questions.

---

### Official Review · Reviewer_CTuq · 2026-03-26

**Soundness:** 2
**Presentation:** 2
**Significance:** 3
**Originality:** 3
**Overall Recommendation:** 2
**Confidence:** 4

**Summary:**

This paper motivates many of the issues in mechanistic interpretability by comparing them to a similar spurious finding in neuroscience: an instance of dead salmons' brain scans being predictive of emotional state (which the dead salmon clearly did not have). The paper persuasively takes the position that these are analogous issues, and thoroughly documents the issues with mechanistic interpretability as it currently stands. The paper ends with a framework -- which the authors called a proposed "shared language" -- with which to improve the methodology and rigor in interpretability research, so that its findings might be more causally accurate, more useful, and withstand the test of further inquiry.

**Compliance With Llm Reviewing Policy:**

Affirmed.

**Key Questions For Authors:**

1. In the introduction, when explaining why the dead salmon problem occurred, you say "The error arose from a failure to correct for multiple comparisons within the statistical analysis pipeline." I assume you mean the inflation of false positives that is usually corrected by adjusting the significance threshold, for example with a Bonferroni correction? If so, it would be advisable to clarify this explicitly, and explain the problem and solution in greater detail. While neuroscience readers will likely be familiar, many ML readers won't know how such a mistake could have occurred, or how to fix it.

2. Related to 1 above, there is a gap between the dead salmon issue in particular and the mechanistic interpretability issues described. Can you draw this link more clearly? The framing of the paper and the title (which is excellent, in my opinion) warrant more explicitly-drawn links and callbacks to this throughout the work if the narrative is to be as cohesive as possible.

3. Why isn't the solution just more rigorous statistical testing, as it is in the dead salmon example/in neuroscience and cognitive science at large?

4. The paper assumes (and I personally agree) that neuroscience and mechanistic interpretability have much to learn from one another and are perhaps solving analogous problems in humans and machines respectively. However, this is something that most readers might not realize or acknowledge. Could you motivate this at a high level?

**Limitations:**

The authors are forthright about limitations throughout the work, even as they take a definite position on the problem, and start to a solution, of the spurious causation-finding problem in mechanistic interpretability.

**Strengths And Weaknesses:**

This is paper has a strong, extremely well-articulated vision for what is going wrong with the field of mechanistic interpretability, and how to start to fix it. It is one of few ICML submissions with this kind of clarity in thought and writing. It makes a compelling and persuasive point with an extensive study of the background literature, including many of the pitfalls of mechanistic interpretability as it exists today.

The key limitation to this paper is the lack of a clear empirical or theoretical contribution. Although the proposal towards the end of the paper is a good start, it doesn't quite match the standard that technical ICML papers generally achieve. I might suggest to the authors that the position paper route would be an excellent one for this style of paper. The articulation of the problem, after some further development, along with the "shared language" developed in the final pages, presents a compelling and well-research position.

---

> ### Author Rebuttal · Authors · 2026-03-27
>
> We thank the reviewer for the engaged reading. We are particularly encouraged by the assessment that the paper provides "well-articulated vision" and makes a "compelling and persuasive point." We respectfully disagree, however, with the conclusion that the paper lacks a technical contribution, and we find the overall score of 2 difficult to reconcile with the strongly positive qualitative evaluation. In particular, if the paper is indeed “one of few ICML submissions with this level of clarity in thought and writing,” we believe this in itself indicates that it makes a meaningful and valuable contribution to the program of ICML.
>
> ## Submitting elsewhere as position
> Transforming the work into a position paper would require changing the framing and argumentation of the work, which the reviewer appreciated. Furthermore, we would like to clarify that while our work does not neatly fit the standard problem/solution/experiment paradigm, it also does not correspond to a position paper. In particular, we do not simply argue that better statistical or causal practices are necessary. The diagnosis we propose is technical, and the framework we introduce as a solution provides a clear roadmap for future work, motivating the field to adopt a similar framing, but itself being a precise and technical formalism. Additionally, although we view this as secondary in the main paper, we do demonstrate empirically one practical consequence of our approach by describing a general framework for hypothesis testing and showing in Figure 4 how it identifies and resolves dead-salmon effects, as well as provides corrected effect sizes for prior works. The framework is not just a conceptual position to guide future work, but already leads to actionable tools that solve dead salmon issues.
>
> ## Dead Salmon originates in multiple hypothesis testing
> Yes, the reviewer is correct. The specific dead salmon issue arised from multiple hypothesis testing without correction, leading to inflated false positives. We will revise the introduction to state this more clearly.
>
> ## Why isn't it enough to do more rigorous statistical testing
> Improved statistical rigor is indeed necessary and not yet consistently practiced, but it is not sufficient on its own.
> Our work shows that striking false positives (dead-salmon-type effects) can emerge from different mechanisms, all linked to non-identifiability. This represents a deeper issue than the original neuroscience example and requires more fundamental changes to our approaches.
>
> First, the objects of interest in interpretability are complex structured entities (e.g., circuits, algorithms, concepts), inferred from highly structured complex systems (e.g., neural activations). How to do rigorous statistical testing is not clear in these settings. This motivates the framework we propose in Section 4, which is designed to accommodate such settings.
>
> Then, we highlight causal overdetermination as a central and non-intuitive issue. This form of non-identifiability implies that even with unlimited access to interventions, we may still observe high false positive rates. We believe this phenomenon has been underexplored not only in interpretability, but also in causality more broadly, and is likely to play a critical role.
>
> Notably, while neuroscience has historically been limited by its ability to observe and intervene on computational units (e.g., neurons), XAI provides a setting where such interventions can be performed systematically. This has revealed the prevalence of overdetermined structures, which we think is a particularly important finding. In this sense, our field may also inform neuroscience, for instance by raising the question of whether and to what extent the brain exhibits similar overdetermination, and how this might affect existing methodologies.
>
> ## More high-level justification for comparison with Neuroscience
> We agree that the motivation for the analogy can be made more explicit at a high level (we currently cite prior work that makes this connection explicit in both AI and neuroscience, e.g., Lindsay 2024). Importantly, the connection is not primarily due to biological inspiration, but because both fields address the same inverse problem: inferring computational structures within a complex system to explain observed behavior. As such, we can draw on philosophical and methodological insights developed in neuroscience to inform the future of XAI. Our work uses the history of failures in neuroscience to open precise questioning on ongoing difficulties in interpretability. We will add a high-level motivating paragraph to make this connection clearer.

---

> > ### Author Rebuttal · Reviewer_CTuq · 2026-03-31
> >
> > Thank you to the authors for their response. The authors state, "In particular, if the paper is indeed “one of few ICML submissions with this level of clarity in thought and writing,” we believe this in itself indicates that it makes a meaningful and valuable contribution to the program of ICML." -- in order to make sure I had this right, I went back and read through the CFP for both the main conference and position paper tracks in detail, and I'm afraid that this is simply not true. I do believe this would make an appropriate submission to the position paper track of this or a future conference, and if the authors (and AC) would like to move this to a position paper submission, I would be happy to revise with this in mind, but in this track, this is paper is not a fit.

---

> > > ### Author Response · Authors · 2026-04-07
> > >
> > > Thank you again for taking the time to engage with our rebuttal and for carefully reconsidering our work.
> > >
> > > We would like to use this final response to clarify why we believe the current work does not fit well within the position track, and why it is appropriate for the main track, even if we agree that it does not follow a standard archetypal structure.
> > >
> > > Our contribution goes beyond advocating for improved statistical practices or highlighting identifiability concerns at a high level. We provide: (i) a concrete technical diagnosis of failure modes in mechanistic interpretability, grounded in non-identifiability and causal overdetermination; (ii) a formal framework that can deal with these issues; and (iii) a practical instantiation of this framework through generalized hypothesis testing against computational alternatives.
> > >
> > > Importantly, this is not just a conceptual statement. We demonstrate empirically that this procedure detects and eliminates dead-salmon effects and provides corrected effect sizes in prior published interpretability results (Appendix A). The framework already yields actionable tools; it is not just a guide for future work. We planned to make this part more important in the main paper (see discussions with other reviewers).
> > >
> > > Should the paper be rejected here, postponed, and rewritten as a position paper, we are concerned that it would require substantially altering its narrative and diminishing its contributions. In particular, it would risk obscuring the fact that we are not merely advocating for a general direction, but committing to a specific technical solution and demonstrating concrete empirical tools. If reframed as a position paper, new reviewers might reasonably ask: what exactly is the position being advocated? whether it is that XAI should focus more on statistical inference, that identifiability is important, or that non-identifiability fundamentally limits interpretability goals. Of course, it would be possible to reshape the paper around one such position, but doing so would significantly narrow both the scope of our argument and the substance of our contributions.
> > >
> > > Conversely, this work could also be reframed to more tightly follow the archetypal technical/empirical paper, for example, by conducting a large-scale empirical audit of prior interpretability methods (probing, SAEs, circuit analyses, etc.) using the hypothesis testing framework, resulting in invalidation of some previous results and revised effect sizes. However, this would shift the focus of the reader toward specific technical choices and empirical coverage, overshadowing the broader unifying perspective and discussion that several reviewers already identified as a key strength. It would also become unnecessarily adversarial to prior work, whereas our goal is to contribute constructively to improving methodological practices. We believe the current version strikes a more appropriate balance by making the problem salient and contextualised from a broad perspective while demonstrating it concretely on representative examples.
> > >
> > > Overall, we believe the paper unfairly suffers from the forced dichotomy between position and technical contributions. Its current structure may make it harder to categorize, but this is also what enables it to contribute meaningfully to ongoing discussions about the limitations of current XAI methodology and how to address them in a principled way through a concrete technical proposal
> > >
> > > As authors, we carefully considered before submission whether this work was better suited for the position track or the main track. We chose the main track precisely to avoid narrowing the work to a single stance, and because we believed the main track to be the appropriate venue for less easily categorised more diverse types of contributions. The position track requires a rigid structure (one more minor example is that the position track requires discussing alternative positions, then what would be a meaningful alternative position to, e.g., the diagnosis that non-identifiable queries leads to high rates of false-positives and non generalisable findings).
> > >
> > > Finally, we note that the other reviewers expressed similar initial concerns regarding positioning, but ultimately supported acceptance in this track, after demanding clearer emphasis on (i) the empirical component and (ii) an explicit limitations section outlining the scope of the current work and the scope of future research. We have discussed with them how we will incorporate the changes in the revision.
> > >
> > > We hope this clarification helps address your concerns.

---

### Decision · Program_Chairs · 2026-04-30

**Decision:**

Reject

**Comment:**

This paper presents a broad statistical-causal critique of contemporary AI interpretability methods, using the "dead salmon" phenomenon from neuroscience as a motivating analogy for false discoveries and non-identifiability in interpretability research. The paper argues that many interpretability methods can be reframed as statistical inference procedures over computational traces, and proposes a unified statistical-causal perspective intended to provide a shared language for reasoning about interpretability failures, uncertainty, and identifiability.

Reviewers generally agreed that the paper is well written, intellectually ambitious, and thought-provoking. The synthesis of failure modes across multiple interpretability paradigms was viewed as insightful, and several reviewers appreciated the attempt to connect interpretability questions with broader issues from statistical and causal inference.

At the same time, substantial discussion centered on the nature and concreteness of the contribution. While the proposed framing is conceptually compelling, several reviewers felt that the work remains primarily organizational and philosophical rather than a sufficiently developed methodological advance for the main conference track. Although the paper sketches a practical framework and includes illustrative empirical examples, many important aspects remain underspecified, including how the proposed formalism should be operationalized in realistic interpretability settings and how explanatory goals and query distributions should be concretely instantiated in practice.

In addition, while the paper presents its perspective as a broad unifying reframing, the relationship to adjacent literature on causal abstraction and abstraction-based interpretability remains underdeveloped. Some of the central ideas, including explanations as surrogate structures evaluated relative to distributions of interventions or queries, appear conceptually related to existing abstraction-oriented perspectives, making the conceptual positioning and novelty of the framework less clearly articulated than desired. The paper would also benefit from deeper engagement with recent work connecting mechanistic interpretability, explanatory faithfulness, and query-preserving causal abstractions, including recent work on causal explanation in mechanistic interpretability (arXiv:2508.11214) as well as recent abstraction-based formulations for reasoning about lossy representations and causal query preservation across levels of abstraction (e.g., arXiv:2509.21607). More broadly, the connection between the proposed framework and established causal hierarchies of explanatory and interventional reasoning could also be articulated more explicitly. While the paper discusses observational, interventional, and counterfactual queries, the relationship between these levels of causal abstraction and the proposed interpretability formalism remains somewhat underdeveloped.

The empirical validation was also viewed as comparatively limited relative to the breadth of the paper’s claims. Much of the contribution is ultimately conceptual, while the practical demonstrations remain relatively narrow in scope and do not yet fully establish the proposed framework as a mature methodology for large-scale interpretability analysis.

Overall, the paper raises important questions and presents a stimulating perspective that is likely to influence future discussions in interpretability research. However, given the current balance between conceptual framing, technical specificity, and empirical substantiation, the contribution was ultimately viewed as insufficiently concrete for acceptance in the main conference track.